# Effects of different amoxicillin treatment durations on microbiome diversity and composition in the gut

Katrine Lekang[1], Sudhanshu Shekhar[2], Dag Berild[3], Fernanda Cristina Petersen[2], Hanne C. Winther-Larsen[1]*

1 Department of Pharmacy, Section for Pharmacology and Pharmaceutical Biosciences, University of Oslo, Oslo, Norway, 2 Faculty of Dentistry, Institute of Oral Biology, University of Oslo, Oslo, Norway, 3 Faculty of Medicine, Department of Infectious Diseases, Institute of Clinical Medicine, University of Oslo, Oslo, Norway

* hannewi@farmasi.uio.no

**Data Availability Statement:** The datasets generated for this study can be found in the Supporting information files ASV_all.csv, metadata_all.csv and taxonomy_table_all_2.csv,

## Abstract

Antibiotics seize an effect on bacterial composition and diversity and have been demonstrated to induce disruptions on gut microbiomes. This may have implications for human health and wellbeing, and an increasing number of studies suggest a link between the gut microbiome and several diseases. Hence, reducing antibiotic treatments may be beneficial for human health status. Further, antimicrobial resistance (AMR) is an increasing global problem that can be counteracted by limiting the usage of antibiotics. Longer antibiotic treatments have been demonstrated to increase the development of AMR. Therefore, shortening of antibiotic treatment durations, provided it is safe for patients, may be one measure to reduce AMR. In this study, the objective was to investigate effects of standard and reduced antibiotic treatment lengths on gut microbiomes using a murine model. Changes in the murine gut microbiome was assessed after using three different treatment durations of amoxicillin (3, 7 or 14 days) as well as a control group not receiving amoxicillin. Fecal samples were collected before and during the whole experiment, until three weeks past end of treatment. These were further subject for 16S rRNA Illumina MiSeq sequencing. Our results demonstrated significant changes in bacterial diversity, richness and evenness during amoxicillin treatment, followed by a reversion in terms of alpha-diversity and abundance of major phyla, after end of treatment. However, a longer restitution time was indicated for mice receiving amoxicillin for 14 days, and phylum Patescibacteria did not fully recover. In addition, an effect on the composition of Firmicutes was indicated to last for at least three weeks in mice treated with amoxicillin for 14 days. Despite an apparently reversion to a close to original state in overall bacterial diversity and richness, the results suggested more durable changes in lower taxonomical levels. We detected several families, genera and ASVs with significantly altered abundance three weeks after exposure to amoxicillin, as well as bacterial taxa that appeared significantly affected by amoxicillin treatment length. This may strengthen the argument for shorter antibiotic treatment regimens to both limit the emergence of antibiotic resistance and risk of gut microbiome disturbance.

and are deposited at NCBI with accession
numbers: PRJNA875126 and SAMN30602556-
SAMN30602715.

**Funding:** This work was funded Department of
Pharmacy, University of Oslo (to KL and HCWL),
https://www.mn.uio.no/farmasi/english/, and
Turning the Tide on Antimicrobial Resistance (TTA)
consortium through Oslo University Hospital
(OUH) (to KL), https://www.ous-research.no/amr/.
The funders had no role in study design, data
collection and analysis, decision to publish, or
preparation of the manuscript.

**Competing interests:** The authors have declared
that no competing interests exist.

# Introduction

More than half of all cells inside the human body are bacteria [1], and most of these reside in the intestine. Numerous different microbial taxa have been identified in the gut system and a the gut flora commonly comprise 800–1000 different bacterial species [2], whereas the majority (> 90%) belongs to the phyla Bacteroidetes and Firmicutes [3]. The gut microbes and their composition may seize a significant effect on our health and wellbeing as they contribute to our metabolism [4,5], nutrient uptake [6], and play a key role in regulating our immune system [7]. A range of factors has been suggested to affect the microbial composition and dynamics of mammalian intestines [reviewed in 8], e.g. birth mode, genetics, gender, hormonal cycle, early life development, infections, diurnal rhythm, medicines, diet and lifestyle. A dysbiosis in the microbial composition or alterations in the abundance of certain taxa, may seize a negative impact on human health and has been linked to the pathogenesis of various diseases, e.g. inflammatory bowel disease (IBD) [9], asthma [10], obesity [11], liver disease [12], diabetes [13], depression [14], cardio vascular disease [15] and Alzheimer [16].

It is well known that simultaneously with eliminating pathogenic target-organisms, antibiotic treatments seize an effect on the microbiome and may disrupt intestinal bacterial homeostasis [17–20]. Broad-spectrum antibiotics such as Ciprofloxacin and β-lactams cause a significant drop in taxonomic richness, diversity and evenness during intake [21,22]. Even though the microbiome to some extent display a resilient capacity and may recover to a composition relatively close to the original, alterations reflected in the abundance of certain taxa, may persist for extensive time [23,24]. Studies involving amoxicillin, have demonstrated a significant shift in the microbiome composition during treatment [24]. A recent review paper focusing on the effect of antibiotics on gut microbiomes conclude that in most of the studies included in the assessment, microbial composition returned to normal within 2–4 weeks after amoxicillin treatment [25]. Short-term effects, such as an increase in Bacteriodetes/Firmicutes ratio, have been described in several papers [18,22], and significant changes in abundance of the phyla Proteobacteria, Actinobacteria, Verrucomicrobia, Deferribacteria, Patescibacteria and Deferribacterota have been observed during treatment and shortly after [18,26]. Nevertheless, it has been suggested that alterations in specific taxa at lower phylogenetic levels, e.g. genus or species, may persist [27]. Despite a reversion to the original state in terms of bacterial phyla diversity, significant alterations in specific taxa can seize an effect on human health and wellbeing [28,29]. Antibiotic induced disturbances in the microbiome have been linked to diseases such as acute or chronic gut infections [30], immune homeostasis [reviewed in 31] and increased risk of allergy and asthma [32]. Hence, it is vital to assess differences on lower taxonomic levels in order to reveal long-persistent changes in the microbiome.

In addition to diversity-related changes in the microbiome, antibiotic treatments stimulate the development of antibiotic resistant bacterial strains [33] and the human gut microbiota has been characterized as a reservoir of antibiotic resistance genes [17]. Infections from antibiotic resistant strains are increasing and are estimated to cause at least 700 000 deaths worldwide each year, and have further been projected to rise to 10 million deaths per year by 2050 [34]. AMR is regarded as one of the most alarming issues for human health, and international institutions are urging to reduce the extensive use of antibiotics [35], which is the main driving force for the development of AMR [33]. Longer antibiotic treatments have been associated with increased rates of resistance [36,37]. In a study comparing 8 and 15 days antibiotic treatment, multi-resistant pathogens was detected with a lower frequency in patients who had received antibiotics for 8 days compared to 15 [36]. While several studies have demonstrated a higher risk of resistance development with longer treatment lengths, there has to our knowledge been no clinical studies comparing treatment lengths, demonstrating increased risk of

AMR development among patients receiving treatments with shorter treatment length. Further, several studies have demonstrated that for the treatment of infections in e.g. the urinary tract, tonsillitis or lungs, reduced antibiotic treatment durations is not inferior to standard treatment duration for patient recovery outcomes [36,38–42]. Therefore, implementation of shorter treatments is in several cases regarded as safe and has been recommended as a standard procedure for certain infections [43,44].

Since several recommendations urge to reduce antibiotic treatment lengths as a measure to limit AMR bacterial strains, it is highly relevant to investigate whether shorter antibiotic courses also reduce the potential dysbiosis effect on the gut microbiome. This may strengthen the argument of reducing treatment durations. In this study, we investigated gut microbiome effects in mice, using high throughput sequencing, after antibiotic treatments amoxicillin with different duration (3, 7 or 14 days). Amoxicillin is a β-lactam antibiotic that targets a moderate range of Gram negative and Gram-positive bacteria. It is a widely used antibiotic worldwide and is commonly used for treating disease such as Streptococcus tonsillitis, sinusitis and pneumonia [45]. In this study, we investigated changes both during and after the intake of amoxicillin, at higher and lower taxonomical levels, to assess whether treatment-related changes are prone to reversion after ending the amoxicillin treatment, and if longer treatments increase the risk of longer lasting changes.

## Materials and methods

### Experimental design

This study was approved by the Norwegian Food Safety Authority (FOTS ID 16504) and was conducted at the Section for comparative medicine, University of Oslo. In total 20 CD1 mice, purchased from Scanbur, were used in the experiment. The mice were eight weeks upon arrival, and all were females. They were kept in quarantine for two weeks prior to the experiment. All mice were housed with bedding and wood material/carton for activity. Cage changes were performed twice a week and they had a cycle of 12 h of light and 12 h of darkness.

The mice were randomly organized into four groups (A, B, C and D), with five mice in each group ($n_{total}$ = 20). The groups were kept in different cages. To distinguish between the replicates throughout the experiment, the mice were ear labeled. Three of the groups (B, C and D) were given amoxicillin added to the water (200 μg/ml). The concentration was based on a previous estimation that mice drink approximately 15 ml per 100 g body weight [46] and weighs approximately 25 grams = 32 mg amoxicillin/kg/day. This is comparable to concentrations commonly used to treat humans (15–35 mg/kg/day) [47]. The water bottle was wrapped in tin foil to protect from light and avoid breakdown of amoxicillin. To mimic natural environmental conditions, the cage environment/feeding was not sterile. Instead, a control group not receiving amoxicillin, Group A, was included. Since amoxicillin have a bitterness in taste, a version with fruit flavor (lemon- and strawberry essence) added to it, was used to make sure that the mice did not recent the water. Group B received amoxicillin for 3 days, group C for 7 days and group D for 14 days. The water levels were closely monitored by daily pen-marks on the bottles, to make sure that the mice were drinking as expected during the whole period of amoxicillin intake. After finalizing the treatment, mice from group B, C and D were housed for 22 days, while mice from group A were housed throughout the whole experiment (Fig 1).

### Sample preparations and sequencing

Fecal samples were collected every fourth day (Fig 1). Sampling was performed by transferring each mouse individually to a clean cage wiped with 70% ethanol. A minimum of five fecal pellets were obtained from each mouse. The pellets were transferred to individual 1,5 ml

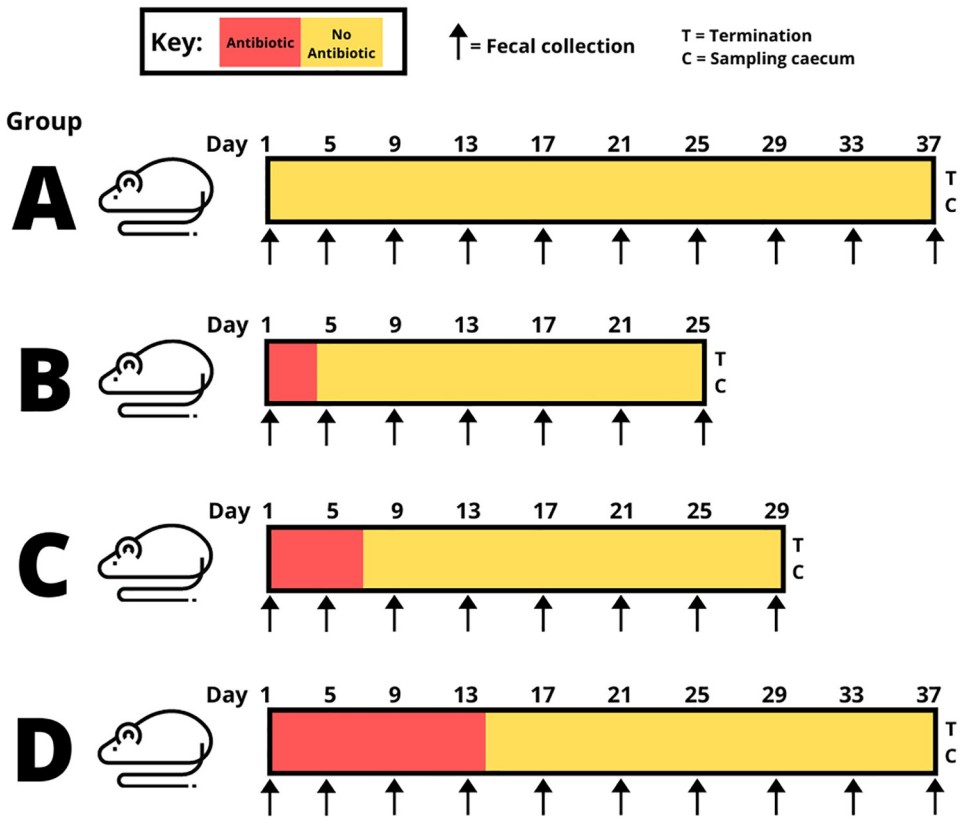

**Fig 1. Experimental set-up to investigate the effect of amoxicillin treatment durations on gut microbiomes.** Four different treatment groups of mice were used. Group A did not receive any antibiotic (control group), group B = 3 days, group C = 7 days, group D = 14 days of antibiotic treatment, respectively. Fecal pellets were sampled from all five replicate mice in each of the four treatment groups, every fourth day (indicated by arrows), and until three weeks after intake of antibiotics had ended.

Eppendorf tubes, and 1 ml RNA*later*™ (Thermo Fisher, United Sates) was added to the tube. The samples were frozen at -80˚C until extraction. At the final experimental day for each group, the mice were euthanized by $CO_2$.

Prior to extraction, RNA later was removed by centrifugation (3 min, 10 000 X g, room-temperature). Genomic DNA was extracted from 3 replicate fecal pellets per sample, using DNA mini stool kit (Qiagen, Germany), following the manufacturers protocol, but with following modifications: one fecal pellet was added two each extraction tube along with extraction buffer. The samples were vortexed for 10 minutes to homogenize, prior to proceeding with the protocol provided by the manufacturer. Shortly described, DNA from the fecal samples binds specifically to a silica-gel membrane while contaminants and excess compounds pass through. A buffer assures separation of PCR inhibitors from DNA. The intact DNA is eluted from the column after a serial of washing steps. DNA from the three replicate samples were pooled before quantification and analysis. The quality was assessed using a UV5nano nanodrop (Mettler Toledo, Switzerland), and concentration by using Invitrogen dsDNA HS Assay Kit (Fisher Scientific, United States) and Invitrogen Qubit 3 Fluorometer (Fisher Scientific, United States).

All samples from sampling day 1–29 and 37 (n = 165) were selected for sequencing. Extracted DNA was equalized based on concentration measurements from Qubit (Fisher

Scientific, United States). Prior to submission, all samples were run on an 0.8% agarose gel stained with GelRed® Nucleic Acid Gel Stain (Biotium, United States) for quality check. The library preparations for 16S rRNA amplicon sequencing were performed by The Norwegian Sequencing Centre (NCS), following a standardized protocol [48], using universal primers targeting the V3-V4 region of 16S rRNA. Resulting 16S rRNA fragments from the 165 samples were sequenced by Illumina MiSeq v3, which provides approximately 20 M paired end reads with 300 bp length.

### Sequence analysis and multivariable statistics

The sequence files were processed using the dada2 pipeline [49] in R statistical software [50]. Filtering and primer-removal were performed using the *filterAndTrim* function using the parameters maxEE = c(2,6), and truncLen = c(260,239). Further, filtered sequences were clustered into Amplicon Sequence Variants (ASVs) using pseudo-pooling, followed by merging the reads using 12 nt as minimum overlap. Finally, chimeric sequences were removed using the *removeBimeraDenovo* function. Resulting sequences were taxonomically classified using the function *assignTaxonomy* and the Silva database v. 138 [51,52]. Total number of sequences and number of sequences obtained after quality control for the various samples are summarized in S1 Table.

All statistics conduced to assess and compare the samples were done in R statistical software [50] and all plots were made using ggplot2 [53]. The ASV dataset was normalized by rarefying to the sample with the lowest number of ASVs (= 18529 ASVs), using the *rarify_even_depth* function in the Phyloseq package [54]. Alpha diversity indexes were calculated using the *alpha*-function in Vegan [55]. Normal distribution was assessed by D'agostinos test [56] and non-parametric Kruskal-Wallis tests were used to assess differences between the samples alpha diversity [57] and Wilcoxon test for pairwise comparison of samples. Rarefaction analysis was performed based on normalized sequence data from the analysed sample set and are shown in S1 Fig. Venn diagrams were created to assess ASVs overlapping and unique for the treatment groups on day 1 (before intake of amoxicillin), day 5 (during treatment) and 3 weeks past end of treatment, by using the VennDiagram package [58]. Boxplots were created for all phyla (Bacteroides, Firmicutes, Actinobacteria, Cyanobacteria, Deferribacterota, Patescibacteria, Proteobacteria and Verrucomicrobia), with data from day 1, day 5 and three weeks past end of treatment, as well as taxonomical orders/families/genera with significant differential abundance between treatment groups. Beta diversity was assessed by calculating Bray Curtis dissimilarity matrices of the normalized data. Non-metric multidimensional scaling (NMDS) using the *ordinate* function in Phyloseq, permutational ANOVA using *adonis* in Vegan, and Beta dispersion test using the *permutest* in Vegan, were performed using these dissimilarity matrices. Further, we identified ASVs with significant different abundance between samples, using the DEseq2 package in R [59] on the normalized data using $p < 0.05$ as a cut-off value.

## Results

### The effect of amoxicillin treatment length on gut microbiome alpha diversity

The alpha diversity indexes demonstrated a significant reduction in bacterial richness (Chao1) (upper panel, Fig 2) for mice receiving amoxicillin for 7 or 14 days (group C and group D), during antibiotic treatment. In these two treatment groups, an average reduction of respective 38% and 34% in richness was observed between day 1 and 5. The richness increased in both groups after intake of amoxicillin ceased. For group C (7 days amoxicillin), richness remained

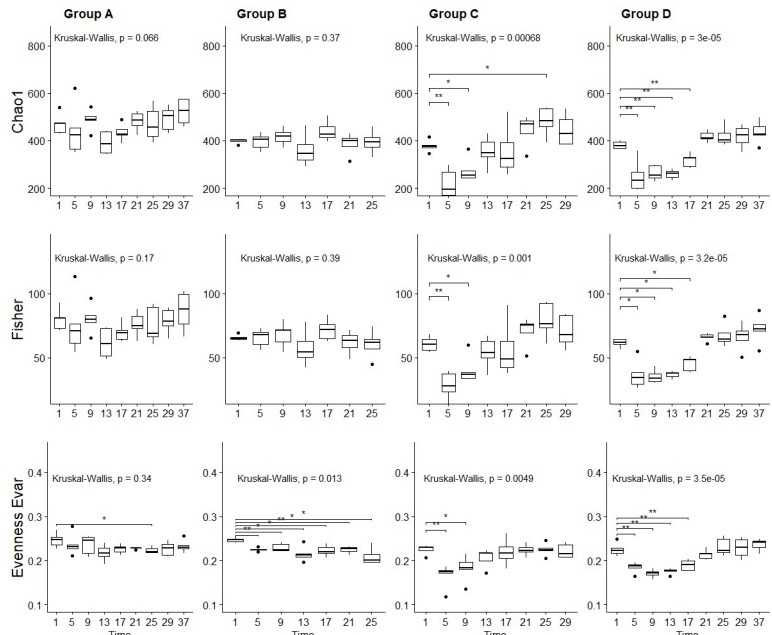

**Fig 2. Alpha diversity results.** Bacterial richness Chao1 (upper panel), Fishers's diversity (middle panel) and Evar evenness (lower panel) indexes calculated for microbiome samples from mice treated with amoxicillin for 0 days (group A), 3 days (group B), 7 days (group C) and 14 days (group D). The x-axis indicates sampling day. Boxes indicate interquartile range (IQR) between the first and third quartiles (25th and 75th percentiles respectively), and the horizontal line inside the box defines the median. Whiskers represent the lowest and highest values within 1.5 times the IQR from the first and third quartiles, respectively. *p*-values to Kruskal–Wallis test is designated on the figure and symbols $^* = p < 0.05$, $^{**} = p < 0.01$, $^{***} = p < 0.001$, according to Wilcoxon test.

significant lower compared to day 1, until day 13, while in group D (14 days amoxicillin) richness remained significantly lower until day 21. Further, Fisher's diversity test was used to investigate diversity (middle panels, Fig 2). No significant changes were detected over time for group A (control) and group B (3 days amoxicillin). For group C and D, a significant reduction in diversity was observed during antibiotic treatment, (Group C: 56% reduction from day 1 to day 5, group D: 40% reduction from day 1 to day 5). The diversity remained significantly lower than baseline (day 1), until day 13 in group C and day 21 in group D (one week past the end of treatment for both groups). In the final days of the experiment there were not detected any significant difference in diversity (Fisher's), when comparing the samples to pre-antibiotic treatment samples from day 1. Further, evenness (Evenness Evar) was significant lower during antibiotic treatment and the following days, compared to day 1, in all the three treatment groups (lower panel, Fig 2), with a reduction of respective 8, 23 and 18% between day 1 and 5, in group B, C and D. During the experiment, there was also detected a significant reduction in evenness in the control group, on day 25, underlining that variation may also occur without the impact of amoxicillin. There was indeed some variation in richness, diversity and evenness among replicas in the control-group. Such variations are expected and are caused by biological differences between the mice. This is elaborated further in the discussion.

In general, the results demonstrated a significant reduction in observed richness, Chao richness, Fishers diversity, evenness and rarity low abundance, during antibiotic treatment (Figs 2 and S2). For most parameters, this appeared to reverse within relatively short time (Fig 2). For treatment groups receiving amoxicillin for 3 and 7 days (Group B and C), no significant difference to other treatment groups was observed one week past end of treatment (S3 Fig).

However, for mice receiving amoxicillin for 14 days (Group D), a significantly lower diversity compared to mice in the other groups, was detected one week past end of treatment. Two weeks past treatment, there were no statistical difference between group D and the control group.

### Taxonomic composition and Beta diversity

Based on the ASVs, we identified the common core microbiome by investigating the overlapping areas in Venn diagrams (Fig 3). The results demonstrated minor differences in the common core microbiome of day 1 (before treatment) and 3 weeks past end of antibiotic treatments, where 36% of the ASVs were affiliated with the common core microbiome on day 1 (420 out of 1177 ASVs), and 34% three weeks past end of treatment (501 out of 1473 ASVs). In terms of the variable genome, Group D (14 days amoxicillin treatment) showed the most notable change, with an increase from 9% of the ASVs identified as unique for group D on day 1 (110 out of 1177 ASVs), to 15% three weeks past end of treatment (227 out of 1473 ASVs). Further, the results demonstrated a reduction in both the number of ASVs affiliated with the common core microbiome and the variable microbiome in group C (7 days amoxicillin) and group D (14 days amoxicillin), during antibiotic treatment. The ASVs common for group A (control), C and D on day 1 of the experiment comprised 37% (436 out of 1177 ASVs), while on day 5 (during antibiotic treatment for group C and D) this was reduced to 28% (327 out of 1180 ASVs). The number of ASVs unique for group A, increased from 11% on day 1 (126 out of 1177 ASVs) to 44% on day 5 (525 out of 1180 ASVs). Treatment group B was not included in the Venn-diagram for day 5 (during treatment), since intake of amoxicillin in this group had ceased two days prior to sampling day 5, hence the samples were regarded as post-treatment samples at this point.

Changes in composition of bacterial phyla, during and after antibiotic treatments were investigated (Fig 4). Bacteriodetes and Firmicutes were the two most abundant phyla in all samples (S4 Fig). The average relative abundance of Bacteriodetes increased during treatment

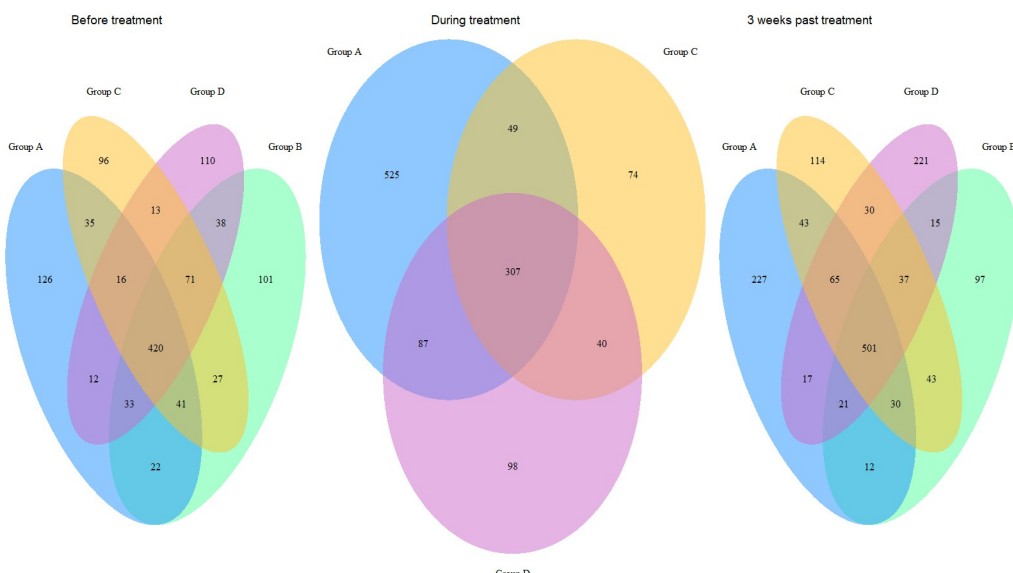

**Fig 3. Venn diagrams demonstrating shared and unique ASVs among mice treated with amoxicillin for 0 days (control group A), 3 days (group B), 7 days (group C) and 14 days (group D), on day 1, during treatment (day 5), and 3 weeks past end of antibiotic treatment.**

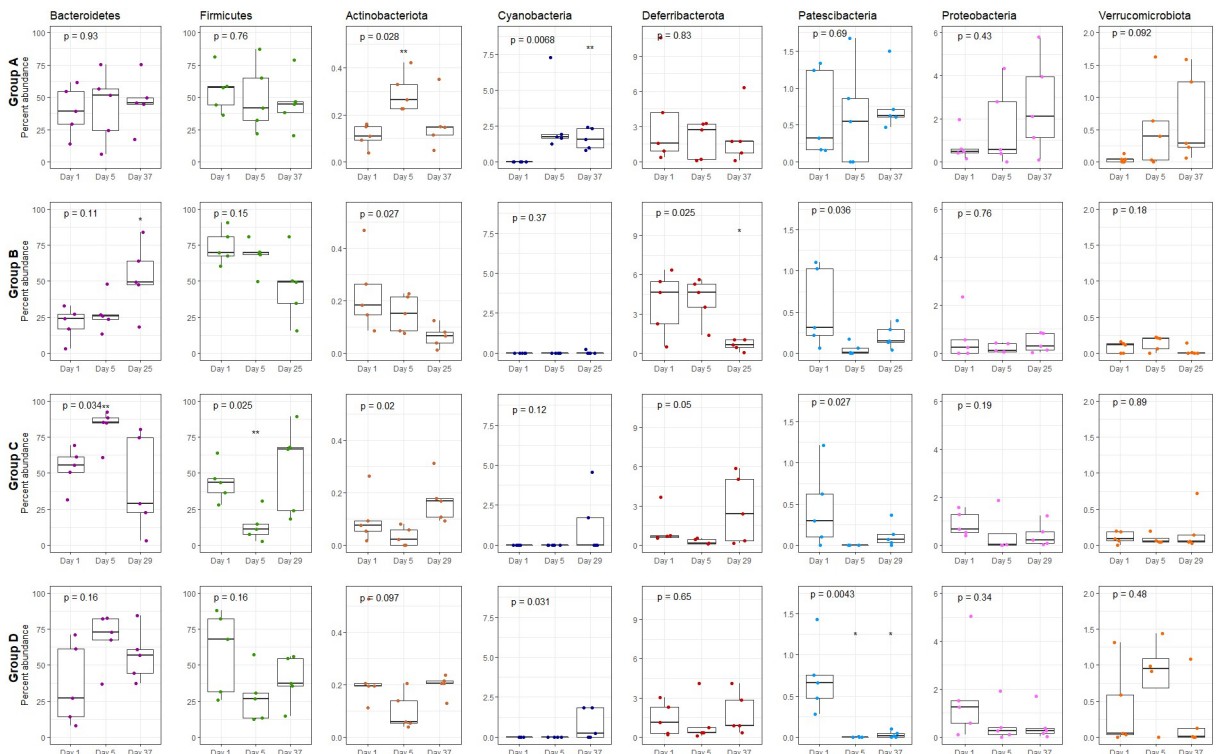

**Fig 4. Abundance of phylum.** Boxplots demonstrating the average percentage of Bacteroidetes, Firmicutes, Actinobacteriota, Cyanobacteria, Deferribacteriota, Patescibacteria, Proteobacteria and Verrucomicrobiota in mice treated with amoxicillin for 0 days (control group A), 3 days (group B), 7 days (group C) and 14 days (group D). Boxes indicate interquartile range (IQR) between the first and third quartiles (25th and 75th percentiles respectively), and the horizontal line inside the box defines the median. Whiskers represent the lowest and highest values within 1.5 times the IQR from the first and third quartiles, respectively. Symbols * = $p < 0.05$, ** = $p < 0.01$, *** = $p < 0.001$, according to Students t-test.

(day 5) in group C and D (7 and 14 days amoxicillin), but was insignificant from day 1, three weeks past end of treatment. In group B (3 days amoxicillin), the average abundance of Bacteriodetes did not change significantly from day 1 to day 5 (two days past intake of amoxicillin). However, it increased between day 5 and day 25. The abundance of Firmicutes appeared to be relatively stable between day 1 and day 5 in group B. However, the abundance was lower at the end of the experiment. In group C and D, the average relative abundance of phylum Firmicutes reduced during antibiotic intake, significantly in group C ($p < 0.01$), while three weeks past end of treatment, the abundance had increased to initial levels.

There were no significant changes in abundance of phyla Actinobacteriota in any of the groups receiving amoxicillin (Fig 4). However, in contrary to the control group (group A), whereas a significant increase in abundance was observed between day 1 and day 5, the average relative abundance decreased in the treatment groups. Cyanobacteria was not detected in any of the treatment groups on day 1 of the experiment. However, further in the experiment, a significant increase was observed in control group A, whereas it increased in all five replicas over time. An increase over time was only detected in two replicas from group C and D and none in group B. Further, there was no consistent trend in the comparative abundance of phyla Deferribacterota between the groups given antibiotics and the control group. However, a significant reduction was observed in group B (3 days amoxicillin) after intake of antibiotics ceased. For phylum Patescibacteria, a reduction in average abundance was demonstrated during antibiotic treatment in all three groups receiving amoxicillin. In mice receiving amoxicillin

for 7 and 14 days (group C and D), this reduction appeared to last ≥ 3 weeks past finalizing the treatment and the relative abundance was significantly lower ($p < 0.05$) in group D. There were no significant alterations in the abundance of Proteobacteria during the experiment. However, while an increasing trend was observed in some of the mice from the control group A, this was not seen in any of the mice from the treatment groups receiving amoxicillin (group B, C, D). Verrucomicrobia increased in abundance in the control group A during the experimental period. This was not detected in group B or C, however for some individual samples in group D, an increase in abundance was observed during treatment (day 5).

Ordination analyses, based on ASV composition (NMDS), demonstrated one distinct cluster of samples from mice receiving amoxicillin (Fig 5a). Interestingly, samples from group B (3 days amoxicillin treatment), collected two days after intake of antibiotics, were not part of this cluster. On the contrary, samples from group C (7 days amoxicillin treatment), collected two days after intake of antibiotics clustered along with the samples collected during antibiotic treatment. A significant difference was verified using a PERMANOVA test ($p < 0.001$). I addition, ordination analyses were run for all samples from the four treatment groups (S5 Fig). No distinct clusters were observed for the samples in control group A (S5a Fig) and PERMANOVA confirmed that there was no significant difference between samples over time ($p = 0.096$). For group B (3 days amoxicillin) (S5b Fig), most samples taken on day 5 and 9 created a separate cluster, and PERMANOVA confirmed a significant change over time

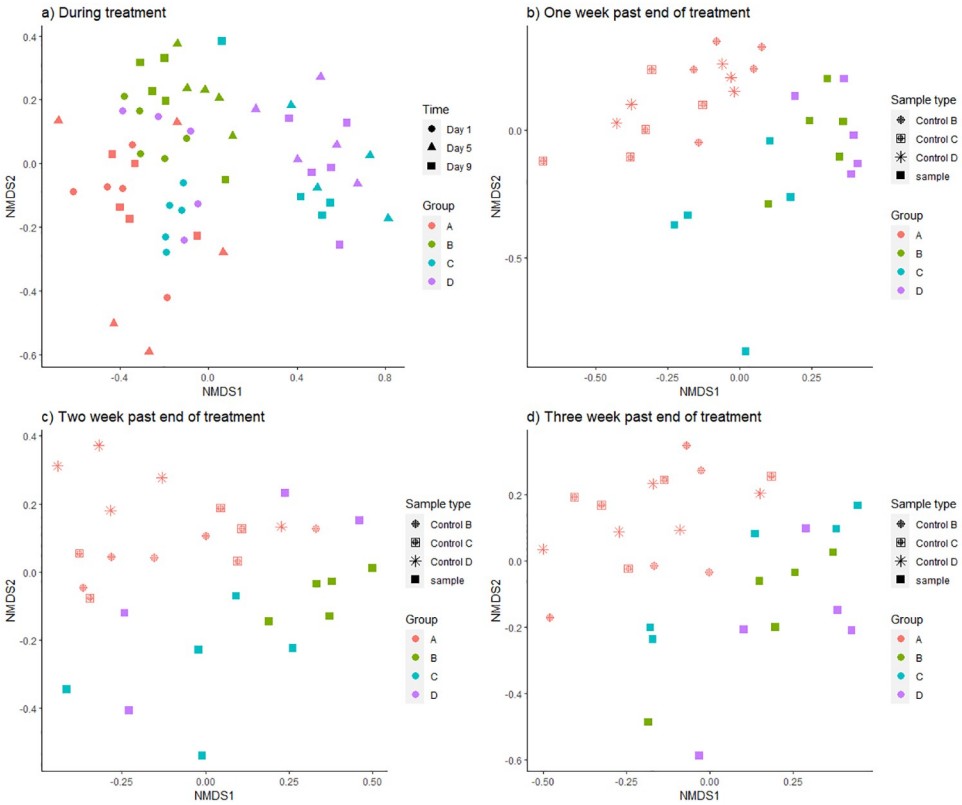

**Fig 5. Beta diversity.** Non-metric multidimensional scaling (NMDS) ordination plot based on Bray-Curtis dissimilarity of community composition, of samples from mice receiving amoxicillin for 0 days (control group A), 3 days (group B), 7 days (group C) and 14 days (group D). **a)** during treatment (day 1–9 of the experiment), **b)** one week past end of treatment, **c)** two weeks past end of treatment, and **d)** three weeks past end of treatment. Colour indicate treatment group, and shape indicate sampling day (a) or treatment group/control group (b-d).

(p < 0.001). This was also the case for group C (7 days amoxicillin) (S5c Fig), PERMANOVA: p = 0.003. For group D (14 days amoxicillin), samples from day 5, 9, 13 and 17 created a distinct cluster (S5d Fig). PERMANOVA confirmed a significant difference between the samples (p < 0.001).

Further ordination analyses, including samples taken 1, 2 and 3 weeks after antibiotic intake, were also performed (Fig 5b–5d). One week after end of antibiotic treatment, samples from all 3 treatment groups receiving amoxicillin (group B, C, D) created a separate cluster from the control samples from control group A which did not receive antibiotics (Fig 5b). The difference was confirmed by PERMANOVA (p < 0.001). This distinct clustering was less distinct 2 and 3 weeks after antibiotic intake ended (Fig 5c and 5d). This was confirmed by PERMANOVA, with p = 0.068 (2 weeks past) and p = 0.277 (3 weeks past). Ordination analyses including only ASVs affiliated with phylum Firmicutes and Bacteriodetes demonstrated the same trend (S6 Fig, with a significant clustering 1 week after end of treatment (p < 0.001 and p = 0.002 for Firmicutes and Batceriodetes respectively) confirmed by PERMANOVA. For Firmicutes, this was still significant 2 weeks after end of treatment (p = 0.015), as well as 3 weeks after end of treatment (p = 0.049). While PERMANOVA did not demonstrate a significant clustering of Bacteriodetes (p = 0.11 and p = 0.59), the permutational beta dispersion test suggested a significant difference between group D and the treatment groups receiving shorter treatments (group B and C), as well as the control group (p < 0.05).

## Differential abundance and indicator taxa

A significant differential abundance between treatment groups was identified for several taxonomic groups three weeks past end of amoxicillin treatment (Fig 6). Among these were two

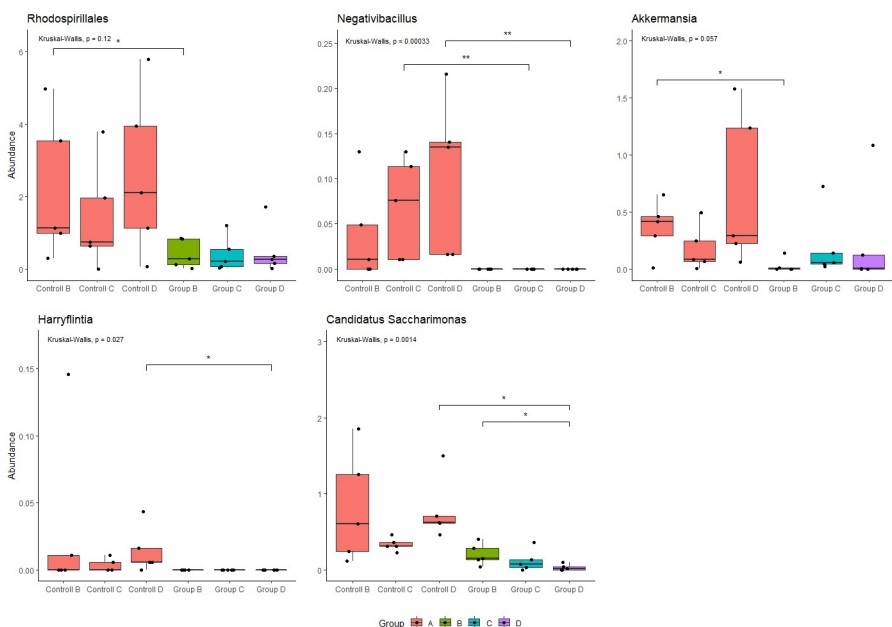

**Fig 6. Boxplots demonstrating the average relative abundance of Rhodospiralles, *Negativibacillus*, *Akkermansia*, *Harryflintia* and Canidatus *Saccharimonas* in mice treated with amoxicillin for 0 days (control group A), 3 days (group B), 7 days (group C) and 14 days (group D).** Boxes indicate interquartile range (IQR) between the first and third quartiles (25th and 75th percentiles respectively), and the horizontal line inside the box defines the median. Whiskers represent the lowest and highest values within 1.5 times the IQR from the first and third quartiles, respectively. Symbols * = p < 0.05, ** = p < 0.01, *** = p < 0.001, according to Wilcoxon test.

genera (*Negativibacillus* and *Harryflintia*) within the Clostridia family *Ruminococcaceae*. Interestingly, neither of these were detected in any of the mice that had received antibiotics for either short or long treatment lengths. A lower abundance of the Alphaproteobacteria order, Rhodospiralles, was also observed in samples from treated mice, with a statistically significant difference between Group B and the control group ($p < 0.05$). This was also the case for the genera *Akkermansia*. Overall, candidate genus *Saccharimonas* also appeared less abundant in mice, which had received antibiotics, with a significant difference between mice receiving antibiotics for 14 days and the related control group ($p < 0.05$). Additionally, the relative abundance of *Candidatus Saccharimonas* was significantly lower in mice receiving antibiotics for 14 days compared to mice receiving antibiotics for 3 days ($p < 0.05$). Fore genus Oscillibacter, there was no significant difference in abundance between amoxicillin treatment groups three weeks past end of treatment. However, in contrary to in control group A and group B, the average abundance of Oscillibacter appeared to increase in the two groups receiving the longest treatments of amoxicillin, group C and D (S7 Fig).

An analysis using DEseq2 in R was used to identify ASVs with a significant higher or lower abundance either in amoxicillin treated/untreated mice, or in mice receiving short/long treatment (Fig 7). We identified 83 ASVs that were significantly associated with amoxicillin exposure during treatment (Fig 7a). Most of them belonged to class Clostridia. *Lachnospiraceae* was the most abundant family (37 ASVs). Four ASVs were annotated to genus *Anaerotruncus*, two

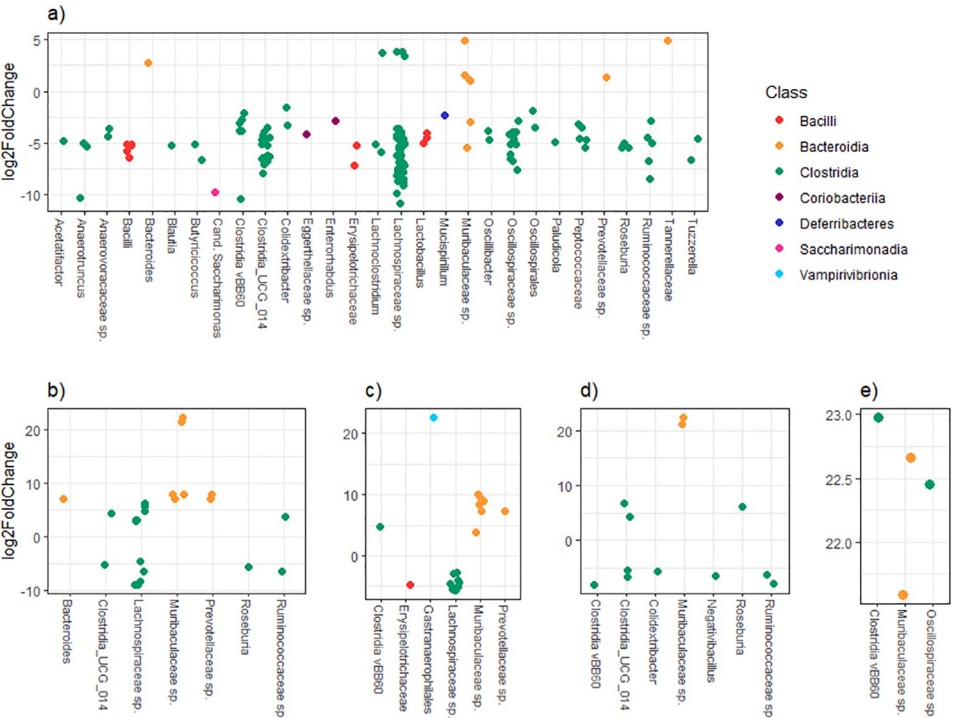

**Fig 7. Differentially abundant ASVs with a significant difference ($p < 0.05$), identified through DESeq2 analysis.** The ASVs are assigned to genus if possible, colour-coded according to their taxonomical class and plotted according to their log2 fold change (indicated on the y-axis), calculated as the levels in samples from the related control group (indicated by * for a-e). **a)** During treatment (treated versus non-treated samples*), **b)** treated versus untreated, 1 week after end of treatment (treatment groups B, C, D versus control group A*), **c)** short versus long treatment, 1 week after end of treatment (treatment groups C and D versus treatment group B*), **d)** treated versus untreated, 3 week after end of treatment (treatment groups B, C, D versus control group A*), **e)** short versus long treatment, 3 week after end of treatment (treatment groups C and D versus treatment group B*).

to *Oscillibacter*, and one to each of the genera *Butyricicoccus*, *Lactobacillus*, *Tuzzerella*, *Paludicola*, *Roseburia* and *Acetatifactor*. Most of the ASVs were less abundant in samples treated with amoxicillin. However, nine ASVs had a significant higher abundance during treatment, among these were six ASVs annotated to class Bacteroidia, whereas four were represented by family *Muribaculaceae*. One week after end of amoxicillin treatment, we identified 24 ASVs, which were significantly associated with amoxicillin treatment (Fig 7b), and 20 ASVs with ≥ 7 days treatment (Fig 7c). All ASVs with a significant lower abundance in treated samples compared to untreated samples, both one and three weeks after end of treatment, affiliated with Clostridia (Fig 7b and 7d). For some of the ASVs, there was also a significant difference between short and long (≥ 7 days) amoxicillin treatment one week after end of treatment (Fig 7c). This was particularly apparent for ASVs from *Lachnospiraceae*, which had a reduced abundance in treated samples, and *Muribaculaceae* which had a higher abundance in treated samples. Further, three weeks after end of amoxicillin treatment, 12 ASVs were identified as significantly associated with amoxicillin treatment (Fig 7d). Three of these affiliated with family *Ruminococcaceae* (ASV 438 was assigned to genus *Negativibacillus*), and one ASV affiliated with *Colidextribacter*. Among the five ASVs with a significantly higher abundance in amoxicillin-treated samples, two was annotated to *Muribaculaceae* within class Bacteroidia. Only four ASVs were identified with a significant different abundance between samples treated with amoxicillin for 3 and 7 days (Fig 7e). All of these were less abundant in samples collected from treatment groups C and D. Two affiliated to family *Muribaculaceae*, and the two others to different lineages of Clostridia.

## Discussion

### Effects on the microbiome during amoxicillin treatment

It is well documented that antibiotics exert a significant impact on the gut microbiome [17,20,25,60–62]. The results from this study confirm this, since a significant reduction in diversity, evenness and richness was demonstrated for samples collected during amoxicillin intake (Figs 2, 3 and S1), as well as bacterial and taxonomic composition (Figs 4, 5 and S3). However, the effect did not appear to accelerate over time with prolonged antibiotic treatment, since there was no significant difference in richness, diversity, nor evenness between day 5 and day 9/day 13 for treatment group D that received amoxicillin for 14 days (Figs 2 and S1). Few other studies investigating antibiotic induced change in the microbiome, include multiple sampling points during treatment as here, but rather assess the pre- and post-treatment effects. However, in a study by Dollive [63], reduced relative abundance of 16S rRNA gene copies was observed in mice receiving amoxicillin for 10 days, from day 1 to day 16, with stable numbers from day 2 to day 16. This, along with the results from our study, suggest an acute effect of amoxicillin, which is not accelerating over time despite continuation of antibiotic intake. The observed effect was further strengthened by ordination analyses, since samples collected during amoxicillin treatment clustered together (Figs 5a, S3c and S3d), rather than creating distinct subclusters.

Previous studies have demonstrated an increasing abundance of Bacteriodetes and a reduction of Firmicutes as an effect of antibiotic treatments (reduced F/B ratio) [reviewed in 20,22,64]. Our results confirmed this since increase in abundance of phyla Bacteriodetes was observed during amoxicillin treatment (Fig 4, group C and D). Our results also suggested a reduction in abundance of phylum Firmicutes (Fig 4, group C and D). Alterations in the F/B ratio have been associated with increased BMI, obesity and metabolic disease [65–67], where a correlation between increased F/B ratio and BMI has been demonstrated.

Among the less abundant phyla, antibiotics have been demonstrated to affect e.g. Proteobacteria and Verrucomicrobia significantly in humans. Phylum Proteobacteria has been reported to increase in human gut microbiomes after antibiotic treatments [18,26]. In contrary to this, our results did not suggest a significant increase of Proteobacteria in the murine gut microbiome, during amoxicillin treatment. This phylum contains a wide range of bacteria and several differences have been identified between the human and murine microbiome [68], which may explain why we did not observe an increase in Proteobacteria during amoxicillin treatment in this study, using a murine model. Phylum Patescibacteria appeared to be among the most affected phylum in the murine gut during amoxicillin treatment since a significant reduction was detected during treatment in group D (Fig 4). A reduction of Patescibacteria has previously been demonstrated during treatment with the antifungal compound fluconazole [69] and it has also been demonstrated as highly sensitive to inorganic contamination in drinking water [70]. However, little information regarding the potential health related effects of these bacterial phyla in the gut is found.

During antibiotic treatment, there was a significant reduction of several Clostridia bacteria, particularly ASVs affiliated with Clostridia UCG-014, *Lachnospiraceae* and *Oscillospiraceae*. A severe effect from antibiotic treatments has been demonstrated on *Lachnospiracea* in previous studies [71]. *Lachnospiraceae* is one of the main producers of short chain fatty acids in the gut and is reckoned to be highly beneficial in regards to human health [72]. In addition, several ASVs affiliated with genera belonging to this family was significantly reduced; *Acetifactor*, *Lachnoclostridium* and *Roseburia*. These are fibre-degrading bacteria, which also are considered positive for gut health. Further, a *Blautia*-classified ASV was demonstrated to decrease during antibiotic treatment. *Blautia* has been described as a genus capable of relieving inflammatory and metabolic diseases, holding antibacterial activity against certain microbial strains [73,74]. Butyrate-producing bacteria are frequently regarded as probiotic and are often found depleted in IBD patients [75]. In our study, we identified reduction in two ASVs affiliated with genus *Butyricoccus* and further five ASVs affiliating with *Ruminococcaceae*, a family known to harbour numerous Butyrate-producing bacteria, during amoxicillin treatment. Several genera within *Lachnospiraceae* do also produce butyric acid, and are regarded as protective against colon cancer. Most of the ASVs detected with a significant differential abundance between samples undergoing treatment and not, was reduced in samples obtained during amoxicillin treatment. However, eleven ASVs affiliating with *Muribaculaceaea*, *Prevotellaeceae*, *Tannerellaceae*, *Lachnospiraceae* and Bacteroides was detected with a higher abundance in mice receiving amoxicillin. *Muribaculaceaea* contribute to propionate production in the gut and its higher relative abundance has been demonstrated to correlate to increased life span in rodents [76]. Further, a negative correlation with obesity- related indicators has also been shown for *Muribaculaceaea* in mice [77]. Family *Prevotellaeceae* comprise several pathogenic strains and a higher abundance of *Prevotellaeceae* has been observed in immunodeficient patients diagnosed with e.g. HIV [78]. Further, it has also been demonstrated that in women with a high genetic risk score for obesity, higher relative abundance of *Prevotellaeceae* increase the risk of obesity [79]. This family has also been shown to significantly increase in patients with IBS, Ulcerative colitis and Chron's disease [80].

## Microbiome effects after antibiotic treatment

Studies suggest that the effect of antibiotics on the human gut microbiome seizes after treatment has stopped, and according to the resilience theory, overall bacterial composition returns close to the original state relatively fast [81]. However, certain changes may last up to months and even years [reviewed in 17], particularly after very long treatments (e.g. > 6 months) [82].

Our results suggested that overall richness and diversity is normalized relatively fast in mice after finalizing amoxicillin treatment (Figs 2, 3 and S2). This is comparable to several previous studies [18,25,83], e.g. in a study assessing changes after 14 days of Vancomycin treatment, whereas diversity and OTU compositional structure was insignificant from baseline 9 days past end of treatment [83]. However, it was also demonstrated that recovery appeared faster with shorter treatment durations. Our results demonstrated no significant difference in diversity between samples collected prior to treatment and two days past antibiotic intake, for the mice in group B (3 days amoxicillin treatment) (Fig 2). However, there was still a significant difference detected two days past treatment for mice in group C (7 days amoxicillin treatment), and three days past treatment for mice in group D (14 days amoxicillin treatment). There was also a significant difference in both richness and diversity, between microbiome samples from mice treated with antibiotics for 14 days and the control group, one week past end of treatment. This was not the case for mice treated with antibiotics for 3 or 7 days, which strengthen the assumption that longer treatments may induce longer lasting effects on the gut microbial diversity. However, in terms of sample evenness, the results demonstrated a longer lasting effect ($\geq$ 3 weeks) for mice receiving antibiotics for 3 days, compared to those treated for 7–14 days. This suggests that even short treatments may induce long-lasting alterations to the microbiome.

In a previous study treating mice with a combination of amoxicillin, metronidazole and bismuth for 10 days, the taxonomic profiles appeared to be close to pre-treatment state two weeks past treatment [18]. In our study, the taxonomy profiles observed for group C and D (Fig 4) suggest that reversion of the bacterial composition is quickly achieved after finalizing the treatment, and that there is a strong resilience in the microbiome. However, in concordance with a previous study [26], the decline in abundance of phylum Patescibacteria appeared to last $\geq$ 3 weeks past end of treatment and was strongest in mice receiving amoxicillin for 14 days. Phylum Patescibacteria is commonly found in low abundance and is not well described, nor significantly linked to pathogenic conditions in the literature. Interestingly, despite no long-term effect on the abundance of Firmicutes, a significant effect of $\geq$ 7 days amoxicillin on the composition of Firmicutes-affiliated ASVs, was suggested to last for $\geq$ 3 weeks. Phylum Cyanobacteria increased significantly in the control group, while not in the three treatment groups. This suggests a suppressive effect from amoxicillin. Cyanobacteria are blue green algae and its role in the gut microbiome is not well described. However, certain studies indicate a potential neurotoxic effect [84].

Even though the overall bacterial diversity and composition in the gut microbiome appear to re-establish relatively soon, certain specific alterations may as already mentioned, persist for longer time [19]. Our results demonstrated that three weeks past end of treatment, several taxonomic groups remained significantly altered in the gut of amoxicillin treated mice (Fig 6). *Akkermansia* appeared lower in all amoxicillin-treated mice compared to the control-group. The difference was statistically significant between group B and it's control group. *Akkermansia* has been suggested to protect against obesity due to its modulating capacity on glucose metabolism [85] and is considered positive for human gut health. Bacteria within Candidatus *Saccharimonas* are lactate and acetate producers and has been suggested to harbour protective capabilities in the development of allergic asthma [86]. Our results demonstrated a significant lower relative abundance of this genus in mice treated with amoxicillin for two weeks and the control group. Interestingly, the abundance of Candidatus *Saccharimonas* was also significantly higher in mice receiving amoxicillin for only three days, compared to the two-week treatment. Genera *Negativibacilus* and *Harryflintia* remained eliminated in mice exposed to amoxicillin three weeks after end of treatment. There was no statistical difference in abundance of genus *Oscillibacter* between the treatment groups and the control groups on the final

day of the experiment (S5 Fig). However, in mice treated with amoxicillin for 7 and 14 days, the abundance appeared to increase over time (from day one to three weeks past end of treatment), in contrary to mice in the control group and mice treated for only 3 days whereas the abundance appeared more stable or declining over time. *Oscillibacter* has previously been demonstrated to increase with deteriorating kidney function [87].

Our data also demonstrated that several ASVs were significantly altered in abundance both one and three weeks after end of amoxicillin treatment (Fig 7). Five ASVs affiliated with *Muribaculaceae* was significantly higher than the control group one week after end of treatment, and two was still elevated three weeks after. Further, these ASVs was significantly higher in group D (receiving amoxicillin for 14 days), compared to group B and C (receiving amoxicillin for 3 and 7 days respectively). As mentioned in the previous sections report of both a significant positive correlation between *Muribaculaceae* abundance and rodent life span [76], a significant negative correlation between *Muribaculaceae* abundance and obesity [77]. A higher abundance of two ASVs affiliated with family *Prevotellaeceae* was still significantly higher in amoxicillin-treated samples one week after end of treatment, with an elevated abundance in group D. This may indicate a possible link to an increased risk of gut inflammations with longer treatments, since this family has been demonstrated to significantly increase in patients with Ulcerative colitis and Chron's disease.

Some studies have assessed the effect of a combination of amoxicillin and clavulanic acid, a β-lactamase inhibitor that may be combined with amoxicillin to increase the antimicrobial effect. In an experiment using amoxicillin/clavulanic acid for 20 days, profound effects were observed during treatment, but only minor one month past end of treatment [88]. A further study administrating the combination of amoxicillin and clavulanic acid for seven days, which is comparable to treatment group C in the current study, also demonstrated significant shifts in the microbiota family composition during antibiotic intake [89]. However, one week after end of treatment, the microbiota showed induced recovery to the original composition.

## Experimental considerations

We observed variation among replicate samples, also on day 1, prior to antibiotic intake (Figs 2, 4 and 5). This is expected, as other studies have also reported individual differences in bacterial diversity of murine microbiome samples [18,26], and mice do not harbour an identical gut microbiome composition, despite living in highly similar environmental conditions. The gut microbiome is shaped and influenced by a range of factors, e.g. being born by separate mothers, genetics and other biological differences [8]. Hence, studying cause and effect on microbial diversity and composition in the gut is challenging and require sufficient replicates, sampling over time and/or detailed metadata. This is particularly important when studying human gut microbiomes as they may live in highly variable environments, on different diets and under highly diverging exposure to other life-style determinants. In our study, we included five replicate mice in each treatment group. We further sampled relatively frequently over a period of ≤ 37 days. Our results from the beta-dispersion test suggested sufficient homogeneity among the replicate samples.

We observed certain significant changes in taxonomic composition in the control group A (Figs 3 and 4). This is however not surprising, since the gut microbiome is known to change with increasing age, and may also be influenced by a range of other variables, e.g. infection, changes in maintenance routines and hormone cycle. A recent study investigating microbiome effects of antibiotics on mice also reported of alterations in the control group [26]. A further study demonstrated that different housing facilities may affect the gut microbiome [90]. Most studies investigating effects of antibiotics on the microbiome include only a limited number of

temporal sampling points. Our study assessed the changes over time with frequent samplings points. The observed alterations in the control group underlines the importance of including sufficient sampling points over time.

Food additives, both synthetic and natural, have been demonstrated to have an impact on gut microbiomes [91]. In this study, we used amoxicillin with added fruit flavour to avoid mice resenting the water due to bitterness in taste from amoxicillin. It is possible that certain changes observed in the microbiome may be caused by these additives rather than by amoxicillin itself. There are no studies investigating the specific flavouring compounds added to the amoxicillin published. However, studies assessing the effects of amoxicillin on gut microbiomes report of results comparable to several of the results shown in this study [88,89].

The murine gut microbiome has frequently been used as a model for the human gut microbiome system [92–94] and mice have been involved in studies investigating the impact of antibiotics on gut microbiomes [26,30,95,96]. However, it is important to emphasize that there are differences between the two systems [reviewed in 68]. In both the human and the murine gut microbiome, most bacteria belong to the two phyla Firmicutes and Bacteriodetes. However, when assessing the microbial composition at lower taxonomic levels, e.g. family and genus, the differences are more evident, both in terms of taxonomic composition and abundance. This can be exemplified by *Prevotella*, *Faecalibacterium* and *Ruminococcus* which are demonstrated as more abundant in the human microbiome, while *Lactobacillus*, *Alistipes* and *Turicibacter* are more abundance in the murine microbiome [reviewed in 68]. Despite such differences, we believe these results are relevant for assessing changes in parameters such as richness, diversity and evenness, induced by antibiotic treatments of different durations, and to investigate reversion rates of the microbiome after different treatments. Further, changes in specific taxonomic groups may provide indications of how amoxicillin may impact the human gut system and health status, and serve as valuable suggestions for what to assess further in studies involving human samples.

An advantage of using mice in such experiments is the possibility to control the environment and external factors that also may affect bacterial diversity in the gut. This increases the likelihood of detecting alterations induced by antibiotics. A further consideration, which should be taken into account when interpreting the results with the human gut in mind, is that mice live in close contact with both its own faeces and the faeces of mice living in the same cage. This may facilitate reintroduction of microbes and increase reversion rates in terms of bacterial diversity and composition. Further, in this study we used only female mice. This means that potential gender-specific effects on male microbiome composition is not revealed by the data in this study. Studies have demonstrated that certain microbiome related effects may be gender-dependent [97,98], also in terms of antibiotic-induced effects [99]. Investigating such effects would be highly interesting in a further study.

Another factor which should be mentioned is the addition of fruit flavour which was added to the water bottles in cages housing mice receiving amoxicillin to limit the risk of bitter taste and reduced water-intake. During the experiment, we carefully monitored the water-intake and found no difference between treatment groups.

## Conclusion

In this study, we demonstrated a significant change in diversity, richness and evenness in microbiome communities of mice receiving amoxicillin. Alterations lasted for the whole treatment period. The results suggested a reversion of the microbiome in terms of diversity, composition and overall taxonomy. However, a longer restitution time was indicated for the group that received amoxicillin for 14 days. A significantly affected ASV-composition was suggested

for phylum Firmicutes and phylum Patescibacteria never seemed to fully recover. Further, despite that several aspects of the overall diversity and composition appears to be insignificantly different from the original state relatively soon after treatment, alterations in specific taxa, with important functions or potential harmful effects may still occur, and we detected several taxonomic groups (*Rhodospirillales*, *Negativibacillus*, *Akkermansia*, *Harryflintia* and Candidatus *Saccharimonas*), and ASVs (affiliated with several genera and families within Firmicutes and Bacteroides) with significantly altered abundance in the gut samples from amoxicillin treatment groups, three weeks after exposure. Some of the identified taxa have been suggested to impose health implications in previous studies. We also identified significant differences between the treatment groups which was linked to amoxicillin treatment length. This combined with the drastic changes to the microbiome during the whole treatment period, emphasize the importance of reducing antibiotic treatment durations, to limit the induction of negative health implications due to microbiome disturbances. This being considered, our findings may have implications for designing rational antibiotic treatment regimens that limit unnecessary exposure to antibiotics.

## Supporting information

**S1 Fig. Rarefaction analysis based on normalized sequence data from the analysed sample set, demonstrating the number of ASVs detected at different sequencing depths.** (TIF)

**S2 Fig. Changes in diversity during amoxicillin treatment.** Observed number of ASVs, richness (Chao1), diversity (Fisher) and evenness (Evar) and Rarity low abundance indexes observed in mice treated with amoxicillin for 0 days (control group A), 3 days (group B), 7 days (group C) and 14 days (group D), during antibiotic treatment (day 1–9 of the experiment). The x-axis shows the group (A/B/C/D) and sampling-number (1 = day 1, 2 = day 5, 3 = day 9). Boxes indicate interquartile range (IQR) between the first and third quartiles (25th and 75th percentiles respectively), and the horizontal line inside the box defines the median. Whiskers represent the lowest and highest values within 1.5 times the IQR from the first and third quartiles, respectively. *p*-values to Kruskal–Wallis test is designated on the figure and symbols * = $p < 0.05$, ** = $p < 0.01$, *** = $p < 0.001$, according to Wilcoxon test. (TIF)

**S3 Fig. Comparison of diversity metrics between treatment groups after amoxicillin intake.** Comparison of the following indexes: Observed richness, Chao1, Fisher diversity, evenness Evar, Dominance Simpson, Rarity low abundance and dominance core abundance, for mice treated with amoxicillin for 0 days (A), 3 days (B), 7 days (C) and 14 days (D), before intake of amoxicillin (Time = 0), 1 week past end of antibiotic treatment (Time = 1wp), 2 weeks past end of antibiotic treatment (Time = 2wp), 3 weeks past end of antibiotic treatment (Time = 3wp). Colours indicates: Grey = no significant difference, red = significant difference; * = $p < 0.05$, ** = $p < 0.01$). (TIF)

**S4 Fig. Stacked bar plots demonstrating the average percentage (y-axis) of each taxonomic group at bacteria phylum level, in mice treated with amoxicillin for 0 days (control group A), 3 days (group B), 7 days (group C) and 14 days (group D).** Different coloured bars represent different phyla (as listed in the legend) and sampling day is indicated on the x-axis. (TIF)

**S5 Fig. Ordination analysis of all samples within individual treatment groups.** Non-metric multidimensional scaling (NMDS) ordination plot based on Bray-Curtis dissimilarity of community composition, of all samples from mice treated with amoxicillin for **a)** 0 days (control, group A), **b)** 3 days (group B), **c)** 7 days (group C), and **d)** 14 days (group D). The colour of the dots indicates sampling day as described in the legend in the diagram, and samples taken during antibiotic treatment are indicated by increased size.
(TIF)

**S6 Fig. Ordination analysis demonstrating beta diversity for phylum Bacteroidetes and Firmicutes.** Non-metric multidimensional scaling (NMDS) ordination plot based on Bray-Curtis dissimilarity of Firmicutes and Bacteriodetes community composition, including samples collected from all four treatment groups, 1, 2 and 3 weeks after end of treatment with amoxicillin. The colour of the dots indicates treatment group (red = control group A, green = group B receiving amoxicillin for 3 days, blue = group C receiving amoxicillin for 7 days, group D receiving amoxicillin for 14 days).
(TIF)

**S7 Fig. Boxplots demonstrating the average relative abundance of *Oscillibacter* on day one (purple) and three weeks past end on treatment (yellow), in mice treated with amoxicillin for 0 days (control group A), 3 days (group B), 7 days (group C) and 14 days (group D).** Boxes indicate interquartile range (IQR) between the first and third quartiles ($25^{th}$ and $75^{th}$ percentiles respectively), and the horizontal line inside the box defines the median. Whiskers represent the lowest and highest values within 1.5 times the IQR from the first and third quartiles, respectively.
(TIF)

**S1 Table. Total number of sequences obtained and number of sequences after sequencing quality control for all samples.**
(DOCX)

**S1 File.**
(CSV)

**S2 File.**
(CSV)

**S3 File.**
(CSV)

## Author Contributions

**Conceptualization:** Sudhanshu Shekhar, Dag Berild, Fernanda Cristina Petersen, Hanne C. Winther-Larsen.

**Data curation:** Katrine Lekang.

**Formal analysis:** Katrine Lekang, Sudhanshu Shekhar, Fernanda Cristina Petersen.

**Funding acquisition:** Katrine Lekang, Dag Berild, Hanne C. Winther-Larsen.

**Investigation:** Katrine Lekang, Sudhanshu Shekhar, Dag Berild, Fernanda Cristina Petersen, Hanne C. Winther-Larsen.

**Methodology:** Katrine Lekang, Sudhanshu Shekhar, Fernanda Cristina Petersen, Hanne C. Winther-Larsen.

**Project administration:** Fernanda Cristina Petersen, Hanne C. Winther-Larsen.

**Resources:** Katrine Lekang, Fernanda Cristina Petersen, Hanne C. Winther-Larsen.

**Supervision:** Katrine Lekang, Dag Berild, Hanne C. Winther-Larsen.

**Validation:** Katrine Lekang, Sudhanshu Shekhar, Fernanda Cristina Petersen.

**Visualization:** Katrine Lekang.

**Writing – original draft:** Katrine Lekang, Sudhanshu Shekhar, Dag Berild, Fernanda Cristina Petersen, Hanne C. Winther-Larsen.

**Writing – review & editing:** Katrine Lekang, Sudhanshu Shekhar, Dag Berild, Fernanda Cristina Petersen, Hanne C. Winther-Larsen.

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
