## [Decision Letter · Decision Letter 0]

26 Apr 2022

PONE-D-21-37884Effects of different amoxicillin treatment durations on microbiome diversity and composition in the gutPLOS ONE

Dear Dr. Winther-Larsen,

Thank you for submitting your manuscript to PLOS ONE. After careful consideration, we feel that it has merit but does not fully meet PLOS ONE’s publication criteria as it currently stands. Therefore, we invite you to submit a revised version of the manuscript that addresses the points raised during the review process.

THERE ARE SOME POTENTIALLY FATAL FLAWS INDENTIFIED, ESPECIALLY IN THE EXPERIMENTAL DESIGN. 

We look forward to receiving your revised manuscript.

Kind regards,

Juan J Loor

Academic Editor

PLOS ONE

Journal Requirements:

“This work was funded Department of Pharmacy, University of Oslo (to KL and HCWL), and Turning the Tide on Antimicrobial Resistance (TTA) consortium through Oslo University Hospital (OUH) (to KL). The funders had no role in study design, data collection and analysis, decision to publish, or preparation of the manuscript”

“This work was funded Department of Pharmacy, University of Oslo (to KL and HCWL), https://www.mn.uio.no/farmasi/english/, and Turning the Tide on Antimicrobial Resistance (TTA) consortium through Oslo University Hospital (OUH) (to KL), https://www.ous-research.no/amr/. The funders had no role in study design, data collection and analysis, decision to publish, or preparation of the manuscript”

3. We note that you have stated that you will provide repository information for your data at acceptance. Should your manuscript be accepted for publication, we will hold it until you provide the relevant accession numbers or DOIs necessary to access your data. If you wish to make changes to your Data Availability statement, please describe these changes in your cover letter and we will update your Data Availability statement to reflect the information you provide..

Reviewers' comments:

Reviewer's Responses to Questions

**Comments to the Author**

1. Is the manuscript technically sound, and do the data support the conclusions?

Reviewer #1: No

Reviewer #2: Yes

2. Has the statistical analysis been performed appropriately and rigorously? 

Reviewer #1: No

Reviewer #2: No

3. Have the authors made all data underlying the findings in their manuscript fully available?

Reviewer #1: Yes

Reviewer #2: No

4. Is the manuscript presented in an intelligible fashion and written in standard English?

Reviewer #1: Yes

Reviewer #2: Yes

5. Review Comments to the Author

Reviewer #1: In this manuscript, Lekang and colleagues seek to determine the acute and enduring effects of various levels of antibiotic treatment on composition of the fecal microbiome of mice. While this is an important problem to address and the authors do a good job of detailing changes at many taxonomic levels longitudinally, there are several weaknesses in study design and lack of novelty that render this work currently unsuitable for publication, as outlined in the following major and minor comments:

MAJOR:

1. Is there justification for using all female animals in this study? Given that sex is an important determinant of microbial composition and both male and female patients receive antibiotic treatment, potential interactions with sex as a biological variable should be included in this study.

2. Were any measures taken to prevent coprophagy? This poses potential confounders including longer administration of the antibiotic than intended (re-administration of malabsorbed antibiotics) and re-inoculation of microbial taxa that may otherwise be eliminated by antibiotic use.

3. Were quantitative or preliminary studies performed to confirm that water consumption (and food consumption, for that matter) was stable across the entire 14 day period? For example, might water consumption decline across the experimental period due to bitter taste, GI symptoms, or changes in microbial composition? These factors would change not only the intended treatment, but microbial composition as well. Additonally, was the fruit flavor added to the water of group A as well? This is not clear.

4. Could baseline samples (prior to antibiotic treatment) be used to normalize inter-subject variability? For example, there is a lot of variability in the alpha diversity values (Figure 2) even at day 1, where assumedly groups B, C, and D should have consumed the same amount of antibiotic. This also raises the point that all groups seemingly experienced increases in alpha diversity throughout the study period, including the control group.

5. The authors do not report the total number of sequences, number of sequences surviving after quality control, and the distribution of sequences among samples. Was rarefaction performed? Alpha rarefaction plots would be important supplementary information that could confirm that sequencing depth was ample for these samples, particularly since the authors highlight findings for low-abundance/rare taxa.

6. The authors highlight the importance of antimicrobial resistance with prolonged antibiotic exposure, but provide no data to support or refute this occurrence. Is any evidence provided regarding function of this altered microbiota? This level of exposure may alter microbial composition, but the less-affected microbiota may simply be shifting to take up the niches of the more-affected microbes, thus not altering function nor causing long-term consequences even if composition remains slightly altered. These are very important and underexplored concepts that would add novelty to a seemingly redundant replication of data already extant in the literature.

MINOR:

1. In the introduction (line 55 and throughout) “intestines” should always be singular (intestine).

2. If the point of the manuscript is to highlight differences in treatment duration on specific microbial taxa, it would be easier to make these comparisons if all treatment groups were graphed together for each taxa of interest. At present, it is difficult to make these comparisons given the number of small graphs that are spatially distant (example – figure 4).

3. A stacked bar graph or pie chart including all taxa (at levels of interest, to highlight impact of antibiotic administration) may more-clearly demonstrate broad effects on microbial composition, even if only supplementary.

Reviewer #2: Overall, the manuscript is well written. I have some comments as the following:

1. Sample size. Please provide power analysis to justify the sample size.

2. On line 164, provide more details regarding the method used for DNA extraction.

3. On line 197, for the usage of DESeq2 to determine differentially abundant taxa. Please provide more details about what read counts were used? And did the author impose any cutoff for the read count, p-value and fold-change?

4. Figures 2-7 are very blurry. Please consider improve the resolution for publication.

5. Please consider depositing the data into public domain.

6. Is there any difference between oral administration and injection in terms of the amoxicillin treatment?

6. PLOS authors have the option to publish the peer review history of their article (what does this mean?). If published, this will include your full peer review and any attached files.

Reviewer #1: No

Reviewer #2: No

---

## [Author Response · Author response to Decision Letter 0]

29 Jun 2022

Dear Juan J Loor, Academic Editor PLOS ONE

Thank you for the review comments that have now been taken into account in the revised version of the manuscript. We feel that the review comments have improved the manuscript. In our response, we have given a point by point answer to all concerns and requests (marked in bold), as well as given the revised text in red italics with line and page numbers to the new version of the manuscript. Modifications of the text are underlined to clearly indicate where changes have been made. We have also included a “marked-up” separate file with track changes where all modifications to the manuscript have been made. 

Reviewer # 1

MAJOR COMMENTS:

1. Is there justification for using all female animals in this study? Given that sex is an important determinant of microbial composition and both male and female patients receive antibiotic treatment, potential interactions with sex as a biological variable should be included in this study. 

In our study we included five mice in each treatment group. All five were female. Hence, any gender-specific effects from amoxicillin on the microbiome will not be revealed. Since this study was meant as a small-scale initial experiment to investigate the effect of different treatment lengths, we chose to include five mice of the same gender (female) rather than five mice of a mixture of female/male mice. By using mice of different genders, we would have risked influence of gender-related effects and reduced statistical certainty. Studies have demonstrated that certain microbiome related effects may be gender-dependent (Bian, Chi et al. 2017, Peng, Xu et al. 2020), also in terms of antibiotic-induced effects (Gao, Shu et al. 2019). Hence, we agree that investigating gender specific effects from different treatment lengths of amoxicillin is highly relevant and would be interesting to proceed in a future study. 

We have now added the following to the discussion in the new version of the manuscript, see line numbers 605-609.

Further, in this study we used only female mice. This means that potential gender-specific effects on male microbiome composition is not revealed by the data in this study. Studies have demonstrated that certain microbiome related effects may be gender-dependent (Bian, Chi et al. 2017, Peng, Xu et al. 2020), also in terms of antibiotic-induced effects (Gao, Jiang et al. 2019). Investigating such effects would be highly interesting in a further study. 

2. Were any measures taken to prevent coprophagy? This poses potential confounders including longer administration of the antibiotic than intended (re-administration of malabsorbed antibiotics) and re-inoculation of microbial taxa that may otherwise be eliminated by antibiotic use. 

These are relevant points that are discussed in line 596-600. While coprophagy can introduce potential confounders, measurements to prevent it can introduce behaviour and physical changes, which can potentially influence the microbiome. The cages were cleaned 2-3 times every week and mice from each treatment group were kept in separate cages. This stated in Material and methods lines 137-138 and 140. Coprophagy could potentially be prevented by use of wired floored cages, however, their use are generally not legal in Norway due to animal welfare legislations. We also highlight that the diversity indexes and most genera returned to initial levels relatively soon after cessation of exposure to amoxicillin through the water, indicating that prolonged effects due to administration are unlike (line 423, 428, 481-487). 

3. Were quantitative or preliminary studies performed to confirm that water consumption (and food consumption, for that matter) was stable across the entire 14 day period? For example, might water consumption decline across the experimental period due to bitter taste, GI symptoms, or changes in microbial composition? These factors would change not only the intended treatment, but microbial composition as well. Additonally, was the fruit flavor added to the water of group A as well? This is not clear. 

No fruit flavour was added to the water for the control group. However, during the experiment we carefully tracked the water levels for each of the four cages (one for each treatment group), and found no difference in drinking habits over time between the treatment groups. General health was investigated regularly during the whole experiment for each individual mouse (weight, fur, clarity eyes) and no signs of diarrhoea were observed in any of the treatment groups. also weighted during the experiment. We have now added the following to the discussion in the new version of the manuscript, see line numbers 605-609.

Another factor which should be mentioned, is the addition of fruit flavour which was added to the water bottles in cages housing mice receiving amoxicillin to limit the risk of bitter taste and reduced water-intake. During the experiment, we carefully monitored the water-intake and found no difference between treatment groups. 

4. Could baseline samples (prior to antibiotic treatment) be used to normalize inter-subject variability? For example, there is a lot of variability in the alpha diversity values (Figure 2) even at day 1, where assumedly groups B, C, and D should have consumed the same amount of antibiotic. This also raises the point that all groups seemingly experienced increases in alpha diversity throughout the study period, including the control group. 

Inter-subject variability is normal in mice and other mammals. There is no standard microbiome common for all mice due to differences related to e.g. genetics, different litter etc (see line 556-561). This is also seen in other studies using mice as model systems for effects on the microbiome (Bazett, Bergeron et al. 2016). However, the inter-subject variability in this study suggests that the number of mice within each treatment group should be increased in future experiments. Further, increased alpha diversity during the experiment was observed for all treatment groups over time. We agree that using baseline samples is a good way to reduce the effect of inter-subject variability. We discuss in line 568-576 possible reasons why alpha diversity increased over time in all groups, including aging. 

5. The authors do not report the total number of sequences, number of sequences surviving after quality control, and the distribution of sequences among samples. Was rarefaction performed? Alpha rarefaction plots would be important supplementary information that could confirm that sequencing depth was ample for these samples, particularly since the authors highlight findings for low-abundance/rare taxa. 

Information on total number of sequences and number of sequences surviving quality control has been added to the supplementary (table S1). Additions are med in the manuscript line 194-195. 

Total number of sequences and number of sequences obtained after quality control for the various samples are summarized in supplementary Table S1.

Rarefaction analyses have been performed and added to the supplementary (Fig S6). Additions are made in the manuscript line 202-203.

Rarefraction analysis was performed based on normalized sequence data from the analysed sample set and are shown in supplementary Fig S1.

6. The authors highlight the importance of antimicrobial resistance with prolonged antibiotic exposure, but provide no data to support or refute this occurrence. Is any evidence provided regarding function of this altered microbiota? This level of exposure may alter microbial composition, but the less-affected microbiota may simply be shifting to take up the niches of the more-affected microbes, thus not altering function nor causing long-term consequences even if composition remains slightly altered. These are very important and underexplored concepts that would add novelty to a seemingly redundant replication of data already extant in the literature. 

Previous studies have demonstrated increased antimicrobial resistance with prolonged antibiotic exposure (Guillemot, Carbon et al. 1998, Chastre, Wolff et al. 2003). Altered functionality would indeed be interesting to assess and is possible with full genomic sequencing (metagenomic sequencing). However, this require more excessive and expensive analysis and was not the main focus of this initial study. 

MINOR COMMENTS:

1. In the introduction (line 55 and throughout) “intestines” should always be singular (intestine). 

We have now corrected this error. See new line 66.

2. If the point of the manuscript is to highlight differences in treatment duration on specific microbial taxa, it would be easier to make these comparisons if all treatment groups were graphed together for each taxa of interest. At present, it is difficult to make these comparisons given the number of small graphs that are spatially distant (example – figure 4)

We thank the reviewer for this comment. However, grouping them together in the same diagram makes the data less accessible to the reader. In our opinion, using the same values on the y-axis makes it possible to compare across treatment groups. 

3. A stacked bar graph or pie chart including all taxa (at levels of interest, to highlight impact of antibiotic administration) may more-clearly demonstrate broad effects on microbial composition, even if only supplementary. 

We have now added these data to the supplementary (Fig S4). 

Reviewer # 2

1. Sample size. Please provide power analysis to justify the sample size. 

Five mice per treatment group does indeed restrict several aspects of the study. However, little information regarding the effect of different antibiotic treatment lengths on the gut microbiome is available and to our knowledge no studies comparable to this has been done. Hence this was meant as an initial study which we hope will provide useful information regarding how to proceed this research. Several previous studies have performed factor-cause effects on murine gut microbiomes, using 4-8 mice (Choo, Kanno et al. 2017, Guida, Turco et al. 2018, Ojima, Gotoh et al. 2020, Liu, Yang et al. 2021, Mao, Xu et al. 2022). 

2. On line 164, provide more details regarding the method used for DNA extraction. 

We have now added more information. See new lines 171-174.

Shortly described, DNA from the fecal samples binds specifically to a silica-gel membrane while contaminants and excess compounds pass through. A buffer assures separation of PCR inhibitors from DNA. The intact DNA is eluted from the column after a serial of washing steps.

3. On line 197, for the usage of DESeq2 to determine differentially abundant taxa. Please provide more details about what read counts were used? And did the author impose any cutoff for the read count, p-value and fold-change? 

We have now added more information. See new line 214.

Further, we identified ASVs with significant different abundance between samples, using the DEseq2 package in R on the normalized data using p < 0.05 as a cut-off value. 

4. Figures 2-7 are very blurry. Please consider improve the resolution for publication. 

We are very sorry that the Fig 2-7 were blurry in the review process of our manuscript as they appeared clear when uploading into the PLOS ONE submission site. This will be carefully taken care of during the potentially publication process. 

5. Please consider depositing the data into public domain.

Upon the acceptance for publication we will deposit the data to Genebank NCBI.

6. Is there any difference between oral administration and injection in terms of the amoxicillin treatment? 

In this study we used mice as a model for potential effects of use of amoxicillin in humans. As amoxicillin is an oral antibiotic it makes little sense to inject the drug to consider the potential effects after injection. When studying the effects of antibiotics on murine gut microbiomes, oral administration is common (Grazul, Kanda et al. 2016)

References

Bazett, M., M.-E. Bergeron and C. K. Haston (2016). "Streptomycin treatment alters the intestinal microbiome, pulmonary T cell profile and airway hyperresponsiveness in a cystic fibrosis mouse model." Scientific Reports 6(1): 19189.

Bian, X., L. Chi, B. Gao, P. Tu, H. Ru and K. Lu (2017). "The artificial sweetener acesulfame potassium affects the gut microbiome and body weight gain in CD-1 mice." PLOS ONE 12(6): e0178426.

Chastre, J., M. Wolff, J. Fagon and et al. (2003). "Comparison of 8 vs 15 days of antibiotic therapy for ventilator-associated pneumonia in adults: A randomized trial." JAMA 290(19): 2588-2598.

Choo, J. M., T. Kanno, N. M. Zain, L. E. Leong, G. C. Abell, J. E. Keeble, K. D. Bruce, A. J. Mason and G. B. Rogers (2017). "Divergent Relationships between Fecal Microbiota and Metabolome following Distinct Antibiotic-Induced Disruptions." mSphere 2(1).

Gao, H., Q. Jiang, H. Ji, J. Ning, C. Li and H. Zheng (2019). "Type 1 diabetes induces cognitive dysfunction in rats associated with alterations of the gut microbiome and metabolomes in serum and hippocampus." Biochimica et Biophysica Acta (BBA) - Molecular Basis of Disease 1865(12): 165541.

Gao, H., Q. Shu, J. Chen, K. Fan, P. Xu, Q. Zhou, C. Li, H. Zheng and P. C. Dorrestein (2019). "Antibiotic Exposure Has Sex-Dependent Effects on the Gut Microbiota and Metabolism of Short-Chain Fatty Acids and Amino Acids in Mice." 4(4): e00048-00019.

Grazul, H., L. L. Kanda and D. Gondek (2016). "Impact of probiotic supplements on microbiome diversity following antibiotic treatment of mice." Gut Microbes 7(2): 101-114.

Guida, F., F. Turco, M. Iannotta, D. De Gregorio, I. Palumbo, G. Sarnelli, A. Furiano, F. Napolitano, S. Boccella, L. Luongo, M. Mazzitelli, A. Usiello, F. De Filippis, F. A. Iannotti, F. Piscitelli, D. Ercolini, V. de Novellis, V. Di Marzo, R. Cuomo and S. Maione (2018). "Antibiotic-induced microbiota perturbation causes gut endocannabinoidome changes, hippocampal neuroglial reorganization and depression in mice." Brain Behav Immun 67: 230-245.

Guillemot, D., C. Carbon, B. Balkau and e. al. (1998). "Low dosage and long treatment duration of β-lactam: risk factors for carriage of penicillin-resistant Streptococcus pneumoniae." JAMA 279(5): 365-370.

Liu, Y., K. Yang, Y. Jia, J. Shi, Z. Tong, D. Fang, B. Yang, C. Su, R. Li, X. Xiao and Z. Wang (2021). "Gut microbiome alterations in high-fat-diet-fed mice are associated with antibiotic tolerance." Nature Microbiology 6(7): 874-884.

Mao, Y.-H., Y. Xu, F. Song, Z.-M. Wang, Y.-H. Li, M. Zhao, F. He, Z. Tian and Y. Yang (2022). "Protective effects of konjac glucomannan on gut microbiome with antibiotic perturbation in mice." Carbohydrate Polymers 290: 119476.

Ojima, M. N., A. Gotoh, H. Takada, T. Odamaki, J.-Z. Xiao, T. Katoh and T. Katayama (2020). "Bifidobacterium bifidum Suppresses Gut Inflammation Caused by Repeated Antibiotic Disturbance Without Recovering Gut Microbiome Diversity in Mice." 11.

Peng, C., X. Xu, Y. Li, X. Li, X. Yang, H. Chen, Y. Zhu, N. Lu and C. He (2020). "Sex-specific association between the gut microbiome and high-fat diet-induced metabolic disorders in mice." Biology of Sex Differences 11(1): 5.

---

## [Decision Letter · Decision Letter 1]

27 Jul 2022

PONE-D-21-37884R1Effects of different amoxicillin treatment durations on microbiome diversity and composition in the gutPLOS ONE

Dear Dr. Winther-Larsen,

Thank you for submitting your manuscript to PLOS ONE. After careful consideration, we feel that it has merit but does not fully meet PLOS ONE’s publication criteria as it currently stands. Therefore, we invite you to submit a revised version of the manuscript that addresses the points raised during the review process.

THERE ARE SOME FINAL MINOR CONCERNS THAT NEED TO BE ADDRESSED.

We look forward to receiving your revised manuscript.

Kind regards,

Juan J Loor

Academic Editor

PLOS ONE

Journal Requirements:

Reviewers' comments:

Reviewer's Responses to Questions

**Comments to the Author**

1. If the authors have adequately addressed your comments raised in a previous round of review and you feel that this manuscript is now acceptable for publication, you may indicate that here to bypass the “Comments to the Author” section, enter your conflict of interest statement in the “Confidential to Editor” section, and submit your "Accept" recommendation.

Reviewer #1: (No Response)

2. Is the manuscript technically sound, and do the data support the conclusions?

Reviewer #1: Partly

3. Has the statistical analysis been performed appropriately and rigorously? 

Reviewer #1: Yes

4. Have the authors made all data underlying the findings in their manuscript fully available?

Reviewer #1: No

5. Is the manuscript presented in an intelligible fashion and written in standard English?

Reviewer #1: Yes

6. Review Comments to the Author

Reviewer #1: Thanks to the authors for addressing the vast majority of my comments, even if only to state them as limitations of the current study and important considerations for future work.

One important piece of information that I could not find in the manuscript that MUST be included prior to publication is whether or not rarefaction was utilized for any/all analyses and to what number of reads all samples were rarefied. I see in supplementary table 1 that there were adequate read depths, but readers will still need to know the methods used for analyses (i.e. rarefaction depth). I do hope that authors follow through with their statement to deposit corresponding data and metadata in a publicly available database.

7. PLOS authors have the option to publish the peer review history of their article (what does this mean?). If published, this will include your full peer review and any attached files.

Reviewer #1: No

---

## [Author Response · Author response to Decision Letter 1]

9 Sep 2022

Dear Juan J Loor, Academic Editor PLOS ONE

Thank you for the review comments that have now been taken into account in this revised version of the manuscript. In our response, we have given a point by point answer to the concerns and requests (marked in bold), as well as given the revised text in red italics with line and page numbers to the new version of the manuscript. We have included a “marked-up” separate file with track changes where all modifications to the manuscript has been made. 

Kind regards

Hanne C. Winther-Larsen.

Response to Reviewer #1.

Reviewer #1: Thanks to the authors for addressing the vast majority of my comments, even if only to state them as limitations of the current study and important considerations for future work.

One important piece of information that I could not find in the manuscript that MUST be included prior to publication is whether or not rarefaction was utilized for any/all analyses and to what number of reads all samples were rarefied. I see in supplementary table 1 that there were adequate read depths, but readers will still need to know the methods used for analyses (i.e. rarefaction depth).

In a previous version of the manuscript we had stated our rarefaction/rarefying analysis as resampling. This information is clarified in line 215-2017. It now reads “The ASV dataset was normalized by rarefying to the sample with the lowest number of ASVs (= 18529 ASVs), using the rarify_even_depth function in the Phyloseq package (ref 54).” 

 I do hope that authors follow through with their statement to deposit corresponding data and metadata in a publicly available database.

The metadata has now been added with the manuscript as three supplementary data files and deposited in NCBI database with access no#: PRJNA875126. This information is added in the manuscript;

Line 627-630:

DATA AVAILABILITY STATEMENT

The datasets generated for this study can be found in the supplementary files ASV_all.csv, metadata_all.csv and taxonomy_table_all_2.csv, and is deposited at NCBI with accession numbers: PRJNA875126 and SAMN30602556-SAMN30602715.

Line 926-927.

Additional files encompass three supplementary tables containing the datasets generated for this study: ASV_all.csv, metadata_all.cvs and taxonomy_table_all_2.csv.

---

## [Editor Report · Decision Letter 2]

22 Sep 2022

Effects of different amoxicillin treatment durations on microbiome diversity and composition in the gut

PONE-D-21-37884R2

Dear Dr. Winther-Larsen,

We’re pleased to inform you that your manuscript has been judged scientifically suitable for publication and will be formally accepted for publication once it meets all outstanding technical requirements.

Kind regards,

Juan J Loor

Academic Editor

PLOS ONE
---

## [Editor Report · Acceptance letter]

19 Oct 2022

PONE-D-21-37884R2 

Effects of different amoxicillin treatment durations on microbiome diversity and composition in the gut 

Dear Dr. Winther-Larsen:

I'm pleased to inform you that your manuscript has been deemed suitable for publication in PLOS ONE. Congratulations! Your manuscript is now with our production department. 

Kind regards, 

on behalf of

Dr. Juan J Loor 

Academic Editor

PLOS ONE